# Enhanced detection of threat materials by dark-field x-ray imaging combined with deep neural networks

T. Partridge [1], A. Astolfo[1,2], S. S. Shankar [3], F. A. Vittoria [1,6], M. Endrizzi [1], S. Arridge[4], T. Riley-Smith[5], I. G. Haig[2], D. Bate[2,1] & A. Olivo [1] ✉

X-ray imaging has been boosted by the introduction of phase-based methods. Detail visibility is enhanced in phase contrast images, and dark-field images are sensitive to inhomogeneities on a length scale below the system's spatial resolution. Here we show that dark-field creates a texture which is characteristic of the imaged material, and that its combination with conventional attenuation leads to an improved discrimination of threat materials. We show that remaining ambiguities can be resolved by exploiting the different energy dependence of the dark-field and attenuation signals. Furthermore, we demonstrate that the dark-field texture is well-suited for identification through machine learning approaches through two proof-of-concept studies. In both cases, application of the same approaches to datasets from which the dark-field images were removed led to a clear degradation in performance. While the small scale of these studies means further research is required, results indicate potential for a combined use of dark-field and deep neural networks in security applications and beyond.

Following pioneering experiments in the mid-60s[1], phase-based X-ray imaging experienced a surge of popularity in the mid-90s, especially due to the wider availability of 3rd generation synchrotron facilities[2]. Although feasibility with micro-focal x-ray sources was already demonstrated in this period[3], implementation with low-brilliance sources in the mid 00' s[4,5] created great interest, as it provided the opportunity for translation into clinical and industrial contexts. Initial focus was on the enhanced contrast of all details in an image enabled by phase effects, either in hybrid images where their highlighting effect is superimposed to standard images[2,3,6,7], or in phase-retrieved images where the two are quantitatively separated, which was again proven first at synchrotrons[8] then with conventional sources[4,9].

Experimentation at synchrotrons using crystals as the phase sensing mechanism[8] then led to the realisation that multiple refraction events caused by object inhomogeneities on a scale below the system's

spatial resolution could also be detected, as they lead to a broadening of the crystal's reflectivity curve[10–12]. By acquiring at least three images of a sample with slightly different orientations of the crystal used to analyse the x-ray direction, attenuation, refraction and dark-field images could be extracted. This marks the birth of multi-modal x-ray imaging: the retrieved attenuation images carry the same information as conventional x-ray images, refraction is the first derivative of the phase change imposed by the object on the x-ray wavefront, which depends on the sample's electron density[3,6], and dark-field depends on the degree of sample inhomogeneity at length-scales below the imaging system's spatial resolution, which was repeatedly proven to provide information complementary to the other two channels[10–13]. Later on, the same ability was transferred to standard labs either using gratings[14] or apertured masks[15]; the latter method, referred to as edge-illumination (EI), is used in this study (Fig. 1).

[1]Department of Medical Physics and Biomedical Engineering, UCL, London WC1E 6BT, UK. [2]Nikon X-Tek Systems Ltd, Tring, Herts HP23 4JX, UK. [3]Nylers Ltd, Marshall House, Middleton Road, Morden, Surrey SM4 6RW, UK. [4]Department of Computer Science, UCL, London WC1E 6BT, UK. [5]XPCI Technology Ltd, The Elms Courtyard, Bromesberrow, Ledbury HR8 1RZ, UK. [6]Present address: ENEA—Radiation Protection Institute, 4 Via Martiri di Monte Sole, 40129 Bologna, Italy. ✉e-mail: a.olivo@ucl.ac.uk

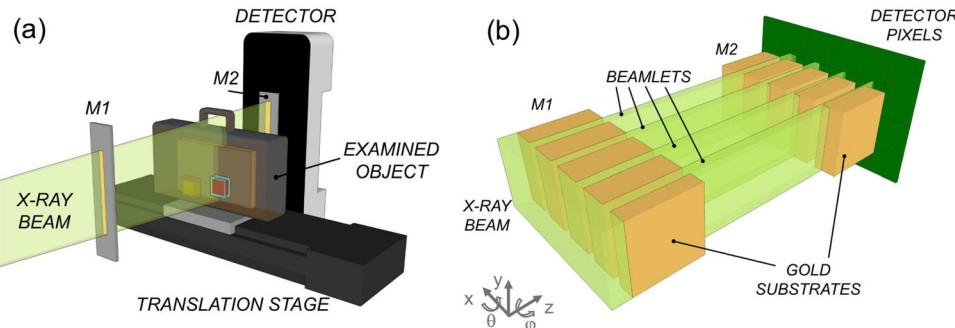

**Fig. 1 | Schematic of edge-illumination x-ray imaging.** This is shown in panel **a**, with a zoom-up on the region between the two x-ray masks in panel **b** (without object). The x-ray beam is split into a plurality of beamlets by a pre-sample mask (M1). These are then interrogated by a second, analyser mask (M2) placed before the detector, which allows assessing their reduction in intensity (attenuation signal), lateral deflection (refraction signal), broadening (dark-field signal).

In EI, the need to acquire at least three input frames (with different displacements of the pre-sample mask, analogous to the re-orientation of the analyser crystal in earlier synchrotron studies) can be avoided if the object is scanned through the beam as in Fig. 1(a), and an asymmetric pre-sample mask[16] is used (see methods).

Pixels in the dark-field image contain the amount of sample-induced broadening suffered by the corresponding beamlet, compared to the case where the sample is absent. As such the pixel-to-pixel variation in the dark-field image provides a texture representing the local variations in the material's microstructure. The textural patterns in dark-field images also make a potential case for discrimination via deep learning[17], especially when the images capture overlapping materials, obfuscating typical micro-structural patterns. Deep neural networks, such as Convolutional Neural Nets (CNNs), are automatic data-driven hierarchical representational techniques, which aim to learn a high-dimensional space over which the input instances can be linearly separated. They are, therefore, suited for applications where data has high diversity, and designing analytical approaches can prove difficult.

An extensive body of work exists in the area of security inspections, with established techniques both in the x-ray and machine learning domains. Prospective advances in x-ray technology are highlighted in an early review[18], and further expanded on in a second one published approximately a decade later[19]. Since these reviews were published, some of the technologies listed therein have reached the commercial stage, namely diffraction and computed tomography (CT). The latter lends itself well to combined implementation with automated detection, as highlighted in a 2015 review[20]. The application of machine learning methods to security-related x-ray imagery has also progressed very significantly over recent years, with an increased use of deep-learning methods following early developments (e.g.,[21]). As well as to baggage, deep-learning methods have been applied also to cargo[22]. This resulted in their inclusion in the US technology roadmap from 2017, with the early commercial systems appearing a couple of years later[23]. However, we note that, even in the context of such an advanced and evolved scenario, dark-field probes matter on a different length scale and on the basis of a different property, namely the material's microstructure, compared to effective atomic number/ electron density for (dual-energy) CT and molecular structure for diffraction[18,19]. As such, it adds an additional, complementary dimension to existing methods, with which it could be integrated. A 2020 review[24] highlights how increasingly complex security scenarios require increasingly advanced combinations of energies, multi-views, and computed vision (which the authors call the 3X strategy); dark-field could add an extra dimension.

Here we show that the textural patterns in dark-field images can be used to improve the distinguishability between materials with similar attenuation but different micro-structural properties, by applying it to the discrimination of threat vs non-threat objects, namely explosives vs a range of benign materials. Prior to the application of deep learning methods, we demonstrate that a combination of multiple x-ray signals allows a better discrimination between explosives and non-threat materials than attenuation alone, and that this is further improved by exploiting the different dependence of phase and attenuation on x-ray energy. We then develop custom CNNs to exploit this increased distinguishability between materials also when they overlap with each other and/or are partially occluded by and included in other objects/materials. We show that our custom-designed CNNs applied to multi-modal, multi-energy X-ray images result in high detection rates between threat and non-threat materials through two pilot experiments: one in which the explosives were occluded by other threat and non-threat materials, and one where they are concealed within various electrical items.

These two experiments should be considered at the proof-of-concept (PoC) level as relatively limited datasets were available for training and testing. Also, a heuristic approach was followed for choosing, building, and optimizing the CNN architectures, which is typical in the deep learning community, due to the limited availability of established theories[25]. We find that the encouraging results we obtained indicate clear potential for the combination of dark-field and CNNs, especially considering that eliminating the dark-field signal from the process leads to a clear degradation in performance (e.g, in the second experiment, a reduction of about 11% in true positives and an increase of about 12% in false positives is observed, see Table 1 below). Although we tested the method in a security context, applications can be envisaged wherever textural differences related to microscopic inhomogeneity are expected, including several areas of materials science as well as possibly to the distinction between diseased and healthy tissues in the life sciences.

## Results
### Material discrimination potential of multi-modal, multi-energy x-ray images
We ran an initial test by scanning a range of acrylic boxes with equal (5 mm) thickness containing four types of explosives (Semtex 1A, Semtex 1H, TSR (i.e., ammonium nitrate), C4—all chosen because they are relatively easy to source and can be handled safely), four non-threat materials with comparable attenuation properties (acrylic, soap, cheese, marzipan), and a stack of paper of equivalent thickness. Results are shown in Fig. 2(a), (b), with the former panel showing average values for each material with their standard deviations, and the latter displaying the content of every image pixel pertaining to a certain material. As can be seen, most materials are effectively separated, some of them only thanks to the additional dark-field channel (e.g., paper vs Semtex 1H), with the only remaining threat/non-threat ambiguity being Semtex 1H vs marzipan. Supplementary Fig. 1 shows that scanned acquisitions do not affect the quantitativeness of the

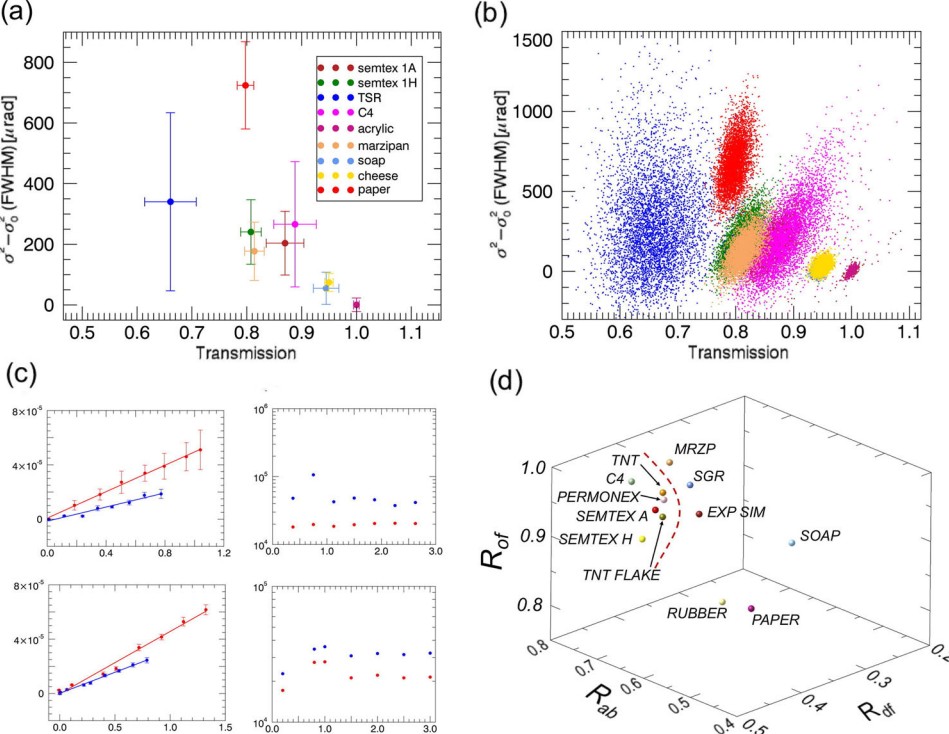

**Fig. 2 | Threat materials discrimination without neural networks.** Top row: scatterplots representing x-ray transmission (normalised to the acrylic box) on the horizontal axis vs dark-field on the vertical one. The latter corresponds to the amount of broadening of the beamlets traversing a certain material caused by the material itself: it is expressed as $\sigma^2 - \sigma_0^2$, where $\sigma^2$ and $\sigma_0^2$ are the squared full widths at half maximum of the beamlets with and without the sample, respectively. Average values with their corresponding standard deviation and values of each individual pixel corresponding to each material are shown in panels **a** and **b**, respectively. The materials' colour coding is the same for the two panels, hence the legend is not repeated in **b** to avoid covering part of the points. The graphs on the left-hand side of panel **c** demonstrate the linearity of the squared dark-field

signal ($\sigma^2 - \sigma_0^2$, vertical axis) vs the attenuation signal ($\mu z$, horizontal axis, with $\mu$ attenuation coefficient and $z$ object thickness) for Semtex H (top) and soap (bottom). Error bars correspond to one standard deviation. The approximately flat behaviour of their ratios as a function of thickness (on the horizontal axis in cm), with the exception of some noise-driven fluctuation at the lowest thickness values, is shown in the graphs on the right-hand side of the same panel. In all graphs of panel **c**, red and blue lines correspond to low and high x-ray energies, respectively. Panel **d** shows how threat materials can be separated from non-threat ones by isolating a certain region of a 3D plot (dashed red line) in which high- and low-energy ratios of $\sigma^2 - \sigma_0^2$, $t$ and $o$ are reported (labelled as $R_{df}$, $R_{ab}$ and $R_{of}$, respectively).

extracted results, by demonstrating that the same values are retrieved from scanned and static acquisitions. It also shows that these are not affected by the scanning speed. The scatterplot of Fig. 2(b) reveals additional information on the microstructure, since for example the values of acrylic are highly concentrated as can be expected from its homogeneous nature, while the only non-industrial material (TSR) shows a significantly larger spread, indicating grain sizes on a much wider range.

Being able to exploit these results in a real-world scenario would require removing three key limitations, namely (1) resolving the remaining ambiguities (e.g Semtex 1H vs marzipan), and moving away from (2) fixed thickness samples and (3) isolated samples. Point 3 requires creating cluttered scenarios where multiple overlapping objects are simultaneously present.

To tackle limitation (1), we used a photon-counting x-ray detector with multiple detection thresholds. This allowed splitting the detected x-ray spectrum in two, and simultaneously acquiring low-energy and high-energy attenuation and dark-field images. Since attenuation and phase effects (to which dark-field belongs) depend differently on x-ray energy, the simultaneous availability of both signals at more than one energy allows for a better material discrimination. Concurrently, we observed that the x-ray masks used were not fully absorbing, and therefore that an additional (offset) image could also be extracted from the scans, by exploiting the (higher energy) x-rays transmitted through the masks septa (see methods). The availability of this plurality of signals also addresses limitation (2), since ratios between different images can be made thickness-independent. For example, the

logarithm of an attenuation image depends linearly on the sample thickness $z$, and dark-field signals can also be linearised against sample thickness[13]; Supplementary Fig. 2 shows that this still holds for dark-field values extracted from scanned acquisitions. The ability to exploit this to obtain thickness-independent values was tested by imaging a range of thicknesses for all considered materials; an example is reported in Fig. 2(c). A variety of signals ($\sigma^2 - \sigma_0^2$, $t$ and offset $o$), which can all be linearised against the sample thickness $z$, is therefore available both above and below the detector energy threshold. While in principle division by one of these signals would be sufficient to make all the others thickness-independent, the resulting ratios would not be linearly independent from each other.

However, plotting ratios of $\sigma^2 - \sigma_0^2$, $t$ and $o$ taken at high and low energies directly on a 3D plot demonstrates an ability to identify a region of space that separates the threat from the non-threat materials (Fig. 2(d)), despite the likely correlation between the $t$ and $o$ ratios. Note that for this result we have used an extended and slightly different set of materials compared to Fig. 2(a), (b): TRS was eliminated, and Permonex, TNT and TNT flake were added to the explosives, while rubber, sugar and an explosive simulant were added to the non-threat ones while cheese and acrylic were removed. Supplementary Fig. 3 shows that data linearisation works for all ratios used in Fig. 2(d).

The results in Fig. 2(d) demonstrate that limitations (1) and (2), namely remaining ambiguities and dependence on sample thickness, have been addressed. Supplementary Fig. 4 shows a possible solution to limitation (3), i.e., material overlap, based on re-normalising the values for every material against the background it is placed on. While

this works, it is a laborious process, which depends on the correct identification of areas occupied only by the overlapping materials. A more practical solution was therefore developed by exploiting the compatibility between the textural nature of the dark-field images and the representational disentanglement offered by deep learning approaches.

### First proof-of-concept CNN test: overlapping threat and non-threat materials

For a first PoC test, we collected approximately 100 multi-modal ($t$, $\sigma^2 - \sigma_0^2$, $o$) images of bags containing an explosive sample of random thickness selected from the 6 materials listed in relation to Fig. 2(d), and another 100 images of bags containing one of the six benign materials listed in the same figure (with cheese replacing the explosive simulant). Various (non-threat) cluttering objects were also placed in the bag for each scan (see methods for details). The aim is to determine whether an input instance of the multi-modal $t$, $\sigma^2 - \sigma_0^2$, $o$ images (horizontally concatenated) contains an explosive or not. Note that we do not aim to recognise the type of explosive, but just to ascertain whether the bag contains an explosive or not. Thus, our deep learning output has only two classes/labels, namely explosives and non-explosives. While, generally speaking, in machine learning classes are mutually exclusive while labels see some mutual overlap, here we use classes/labels interchangeably, as any given two classes may always have some partial overlap owing to some attributes.

We made 10 non-skewed, random 70/30 training/testing splits of the dataset, i.e., in each of the 10 splits, 70% of the images were used for training, while the remaining 30% were used for testing. Due to sample scarcity in the dataset, we did not create a separate validation set, but used k-fold cross-validation during training, with separate normalization on the training and test folds to avoid data leakage. We saw this as the choice least prone to overfitting with our limited datasets (see discussion). We further created 2 random crops for each image (constrained within a limit so that salient portions do not get adversely affected), both for training and testing sets. With 10 random splits, this gave us $70 \times 10 \times 2 \times 2$ (for explosives and non-explosives = 2800 images to train), and $30 \times 10 \times 2 \times 2 = 1200$ images to test with.

We evaluated two of the most widely used CNN architectures, namely GoogleNet[26] and Inception ResNet[27]. Following the standard convention, we configured both nets to output a 640-dimensional output vector at their last layer. Three loss functions have been evaluated, namely softmax, cross-entropy and hinge[28–30] (see methods). As part of an ablation study, we established that hinge loss, for our dataset, always outperforms cross-entropy, and therefore only compared softmax and hinge loss in the ensuing experiments. While cross-entropy is often a more popular choice than hinge loss, the latter is known to give better margins in situations where the labels are very subtle to segregate (e.g., some features are similar across labels, with very few actually separating them)[31].

While in typical CNN training scenarios around 300–400 samples are expected per output class, our training dataset had only around 280 samples (140 each for explosives and non-explosives). In such a scenario, it is prudent to use transfer learning[32,33]. We chose to use ImageNet for this since, owing to its diverse categories, it helps a CNN learn very generic feature representations, which can be used to perform transfer learning for a wide range of smaller datasets.

For both GoogleNet and Inception ResNet, we tried transfer learning with 1, 2, and 3 additional 640-node fully connected layers, and tested all combinations with both the softmax loss and the hinge loss (we will refer to this as type I architecture; see methods for details). We found that Inception ResNet almost always outperforms GoogleNet, which is in line with previous findings[27]. Regarding transfer learning, we got the best results with just one additional fully connected layer. This could be expected since, with an increased number of additional layers, most transfer learning approaches tend to overfit

the training data, due to an increase in the number of possibly redundant parameters. Results obtained with the type I architecture are reported in Fig. 3 (all bars but the last one on the far right).

While this approach provided good results, we observed that it failed where there were subtle differences in the textures of image details, i.e., our model could not differentiate between similar textures in images when they belonged to different categories. To address this, we add texture recognition into the type I architecture, creating our type II architecture (see methods). We trained an Inception ResNet on the Describable Textures Dataset[17], which contains 47 types of textures and is widely used in the computer vision community. We performed this additional training using a cross-entropy loss since, theoretically, a single input image can have components of multiple textures. We then concatenated the 47-dimensional output of this net with the last layer output of the Inception ResNet and trained for our explosives vs non-explosives dataset using the hinge loss, as it performed best in our tests with the type I architecture. We therefore now use a (47 + 640)-dimensional space for separating explosives from non-explosives. As can be seen from the last column of Fig. 3, the texture-based modification improved the performance, leading to 598 true positives, 600 true negatives, 0 false positives and 2 false negatives, corresponding to 99.6% recall, 100% precision and 99.8% accuracy. While encouraging, several caveats must be applied to these results, since the dataset was limited. A key thing to note, however, is that when the above exercise is repeated using a concatenation of $t$ and $o$ only (i.e., eliminating the dark-field image channel $\sigma^2 - \sigma_0^2$), which represents a proxy for the currently employed dual energy methods, the observed accuracy dropped from 99.8% to 93.6%. This clearly indicates the importance of the dark-field channel.

We also note that the best results are obtained without data augmentation, an explanation for which is provided in the discussion. Some example images from this test are shown in Supplementary Fig. 5.

### Second proof-of-concept CNN test: explosives concealed in electrical items

The success of the first PoC CNN test encouraged us to pursue an additional test in an area that would challenge our (and other) testing methods, namely concealing small quantities of an explosive in electrical items. Varying quantities of C4, chosen because it was readily available in sufficient quantities, were concealed in a laptop, a mobile phone, and a hair-drier (see methods for details). Approximately 600 scans were performed, half of electrical items containing concealed C4 and half of electrical items without C4. The electrical items were inserted in a bag also containing a series of additional cluttering items (see methods). To mitigate the effects of the significantly varying thickness values across the bags, ratio images were created by dividing dark-field and attenuation images (see above). This creates an overall flatter image, which however maintains a trace of the C4 texture, which is picked up by the neural network. Figure 4 show examples (in panels (a–f) where a small quantity of C4 was concealed. The small size, alongside the fact that a single energy attenuation image is shown, instead of the typical dual energy one which is colour-coded to separate organic from inorganic materials, explains why the material is hardly visible in the attenuation images of panels (a) and (d). This notwithstanding, the textural signature left by C4 in the ratio images is just about perceptible in panels (c) and (f) (see arrows), which explains why it is picked up effectively by the CNNs; this could provide significant assistance to an operator.

Since this is a different task compared to the previous PoC test (in the sense that some disambiguation across the properties of electric items also needed to be done), a different, split-network[34] CNN architecture was set up. This is a stacked architecture featuring two layers, the first targeted to segment objects from one another, and the second to discriminate the C4 texture inside the segmented objects (see

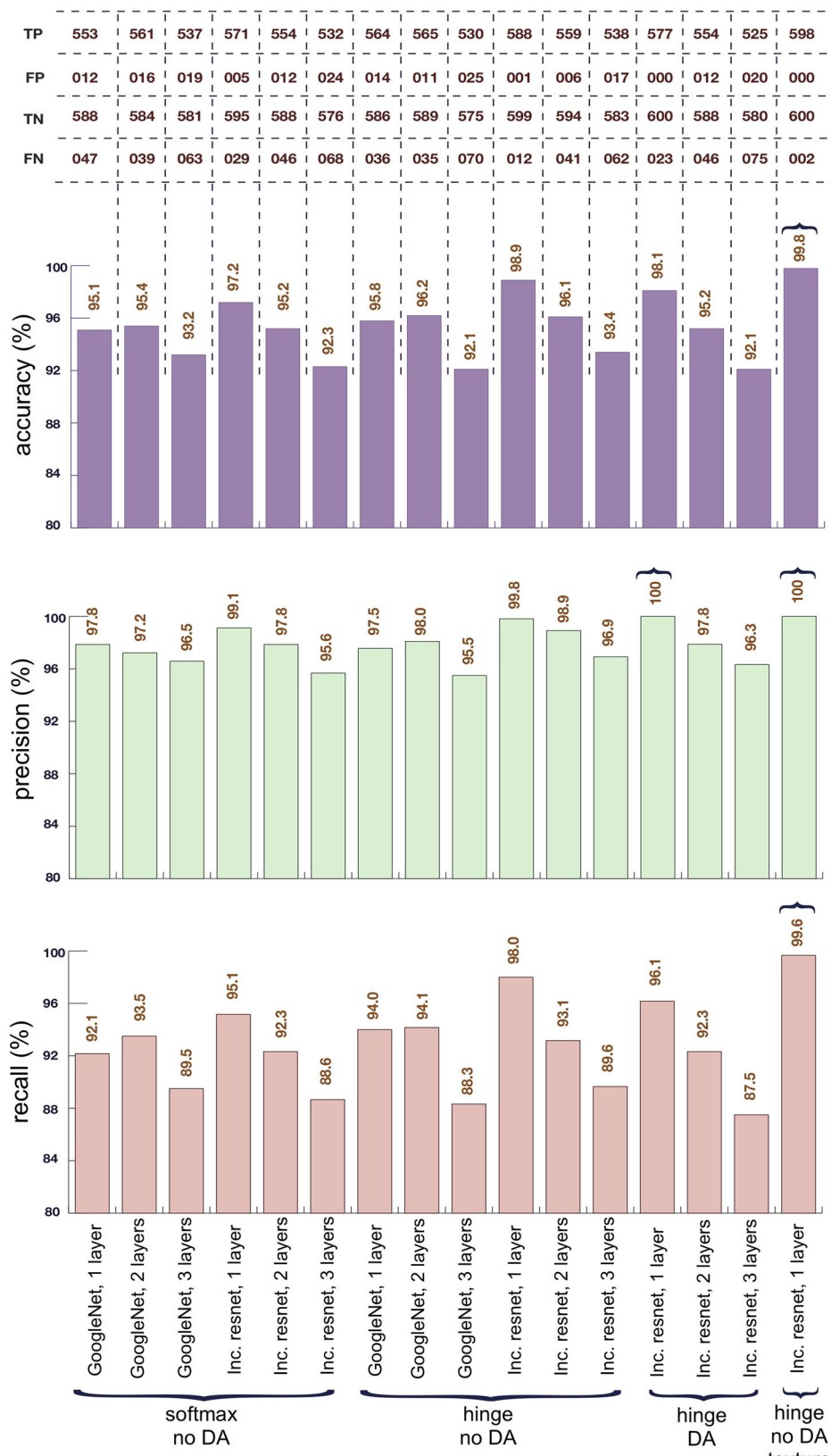

**Fig. 3 | Outcomes obtained using different CNN architectures.** As indicated at the bottom of the figure, examples feature different numbers of connected layers for transfer learning, different loss functions, presence or absence of data augmentation (DA in the figure) and additional training on textures. The non-texture architectures are type I, while the texture-including architecture is type II. We obtained the best performance with the Inception Resnet CNN, no data augmentation, only 1 additional fully connected layer for transfer learning, hinge loss, and fusion of texture recognition (rightmost column). All results are averaged over the random splits we created as a part of our dataset (TP = true positives, TN = true negatives, FP = false positives, FN = false negatives; recall = [TP/(TP + FN)], precision = [TP/(TP + FP)], accuracy = [(TP + TN)/ (TP + TN + FP + FN)]).

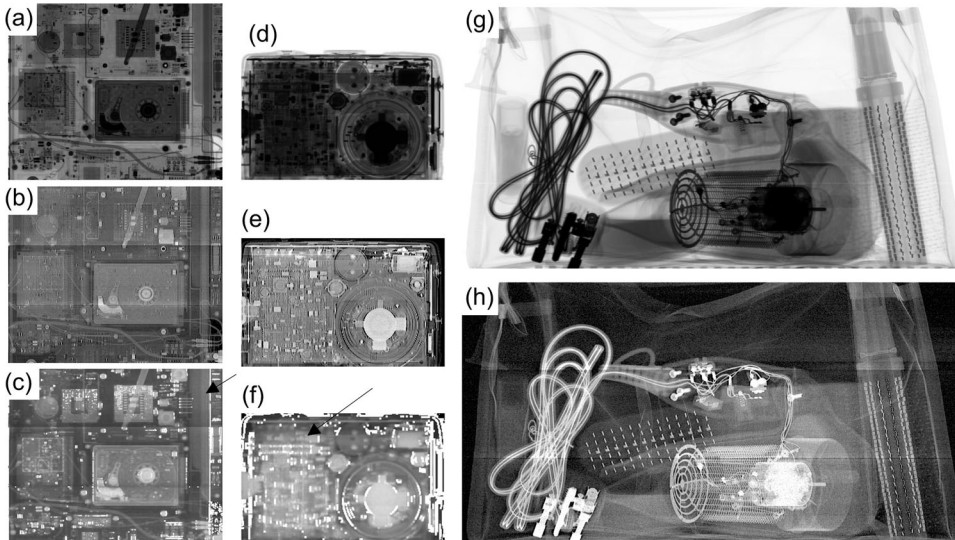

**Fig. 4 | Example images from PoC test 2.** Attenuation (**a**, **d**), dark-field (**b**, **e**) and ratio (**c**, **f**) images of the laptop and mobile phone (respectively) in which a small quantity of C4 was concealed are shown. Arrows in panels **c** and **f** point at the texture C4 leaves in the ratio images. For completeness, panels **g** and **h** show attenuation and dark-field images (respectively) of a cluttered bag containing the hair drier.

methods). This means that, unlike in the first PoC test, the textures of the materials do not get convolved with those of objects surrounding them, making it easier for a single network to disambiguate. Since this approach proved particularly effective in identifying the C4 texture also when this was highly occluded by other objects, further training on textures was not undertaken in this case; this could still be implemented in future applications involving more cluttering items and types of materials. 80 images, half of which with hidden C4, were excluded from the training process and used for the trial. This consisted in showing the images to four trained security officers and running them through our CNN architecture in a formal trial supervised by a member of the funding Authorities. Details on the modality with which the de visu trials were conducted are provided in the methods. Results from the trial are summarised in Table 1.

As can be seen, the CNNs outperform the human operators and, at least in this limited PoC test, achieve a 100% true positive rate. As an additional test, we repeated the process after removing the dark-field images: this resulted in a decrease of about 11% in precision and around 20% in overall accuracy. Once again this highlights the key contribution of the dark-field signal in subtle texture discrimination, and particularly explosive detection. The time taken for the CNN inference with and without dark-field signals is almost identical, since the overall architecture remains same and we simply removed the dark-field images.

To test the advantage provided by the proposed architecture, we processed the same dataset using standard, widely available CNN architectures, namely GoogleNet and Inception ResNet, with the results presented in Supplementary Table 1. These show both the improved performance of the proposed architecture, and that also the performance of standard CNN approaches is boosted by the inclusion of the dark-field signal.

**Table 1 | Results from the de visu trials with security officers, and from the CNN tool**

| Trial | True Positives | False Positives | Average Time |
|---|---|---|---|
| De Visu-security officers | 48.8% | 30.6% | 8 min 45 s |
| CNN (with dark-field signal) | 100% | 17.5% | 1 min 58 s |
| CNN (without dark-field signal) | 89.1% | 29.7% | 1 min 57 s |

Time refers to the analysis of the entire dataset, consisting of 80 images.

The CNN results reported in Table 1 were obtained with a positive (i.e., explosive) classification threshold of 0.5. For completeness, the variation of true positive vs. false positive rates as a function of the positive classification threshold is reported as a Receiver Operating Characteristic (ROC) curve in Supplementary Fig. 6.

## Discussion

The first key result presented in this article is that materials can be effectively discriminated by combining attenuation and dark-field signals, especially if information extracted at more than one x-ray energy is simultaneously used. We show that the effect of material thickness can be eliminated by dividing signals that depend on thickness in the same way; however, the mixing of signals resulting from overlapping materials cannot be eliminated in a two-dimensional, projection-based imaging method. While we show that two overlapping signals can be factored out if an area where one of the two materials is imaged separately is available (Supplementary Fig. 4), this is a laborious process, based on a condition that may not always be verified. We, therefore, explored the use of CNNs as a more practical approach, based on the hypothesis they will be able to pick up the dark-field texture of the materials of interest even when these overlap with others. Our preliminary results show that this seems indeed to be the case; however, several caveats must be made on both PoC CNN tests carried out so far. In the first test, a limited dataset was used, with a single target material present in any given sub-image. In the second test, a single explosive material was used, and operators were only presented with a single attenuation image rather than the dual energy version they are used to seeing. Also in this case, the overall dataset was relatively small.

Another limitation is the heuristic approach through which the PoC CNN architectures were developed. For example, we pre-trained on ImageNET because it is one of the most diverse datasets for general applications, then progressively refined the approach e.g., by adding network branches and incorporating additional training on textural datasets; this was then proven experimentally to improve the detection rates. In future studies, the knowledge acquired through these PoC tests should be used from the beginning in the architecture design[35]. This would require a detailed theoretical analysis, involving finding sample complexity and hypothesis class capacity bounds.

The limited datasets available left little room for ablation studies, notably in PoC Test 1 where we only did k-fold cross-validation. While

k-fold cross-validation is known to reduce overfitting in the presence of limited data and early stopping training, it cannot compare to situations where there is sufficient data to hold-out for testing[36]. However, cross-validation is empirically known to have a comparable effect to held-out data testing situations when the features emanating from the network present a pronounced overlapping across the classes, which happens especially when classes are relatively complex[36]. More data would enable us to test our CNN hypothesis class better and draw a thorough comparison with the efficacy of cross-validation procedures.

We note the better results obtained without data augmentation (rotation, flipping, scaling, altering contrast and illumination properties). While this may look surprising, we believe it could be due to the peculiar nature of the features created by dark-field imaging. These arise mostly from the materials' grainy features, so some operations typical of data augmentation such as scaling may lead to e.g., a variation of the average grain size/distance and therefore to a possible overlap with a different class. It should also be noted that the X-ray imaging system used is sensitive to phase effects in one direction only (due to the use of long slits as mask apertures: this is a common feature of several XPCI methods, including those using gratings and crystals). Therefore, a 90° rotation could lead to an unrealistic texture, which cannot be encountered in the experimental data.

Within all these caveats, the CNN tool showed good performance in both PoC tests, with a high detection rate of the threat objects. In the second test, only false positives, and no false negatives, were observed, which is encouraging as missing explosives is a greater cause for concern than a limited number of false positives. Importantly, in both cases the performance was negatively affected if the process was repeated using only the multi-energy attenuation x-ray images (a proxy for current practice) and excluding the dark-field ones.

A wider, more complex and complete study (more materials, more clutter, thicker bags etc.) should now be pursued in collaboration with the Authorities, with a CNN architecture that can take advantage of all the lessons learnt in this preliminary study. However, we find that the pilot results obtained in this project show promise for a significant improvement of threat detection, for example by providing valuable assistance to the operators. While it may be unrealistic to expect 100% accuracy values in a more realistic, larger-scale study, we note that the (multi-energy) dark-field signal can only add to the discrimination potential of current dual energy, attenuation-only methods, with the textural analysis through DNNs further adding to the detection capabilities.

While here we tested the approach in a security context by applying it to the discrimination between threat and non-threat materials, we expect it to be applicable to a wider range of areas where materials with different microscopic structures need to be discriminated, and possibly identified. This includes several areas of materials science and industrial testing, as well as possibly the distinction between healthy and diseased tissues in medicine: for example, a difference in texture between breast tumours and surrounding healthy tissue was observed already in the very early days of experimentation with dark-field x-ray imaging[37]. We also note that, while here we used EI as the method to extract dark-field signatures, this could equivalently be done with methods based on e.g., gratings[14] (or, at synchrotrons, crystals[10–12]), which expands the applicability of the proposed approach.

## Methods
### Imaging system
The imaging system features an X-Tek 160 X-ray source with a fixed tungsten anode and tuneable focal spot size; this was set to approximately 80 μm for all the experiments described in this article. All reported results were obtained with an accelerating voltage of 80 kVp with the exception of the trial on electrical items (Fig. 4), for which this was increased to 120 kVp to allow for higher penetration.

The detector is an XCounter XC-FLITE FX2 CdTe CMOS-based photon counter, featuring 2048 ×128 square pixels with 100 μm side. The detector features two thresholds that enable discriminating x-ray interactions based on their energy: the first was used to cut-off noise leading to quantum-limited behaviour of the detected photons, and the second to separate them into two (high and low energy) spectral bins. Two masks are used, both fabricated by Creatv Microtech (Rockville, MD) to the authors' design by electroplating approximately 200 μm of gold on 500 μm thick graphite substrates. The detector mask has an area of $1.5 \times 20$ cm² and consists of a series of regularly spaced apertures extending over the 20 cm length, with a period of 97.5 μm and an aperture size of 28 μm. The pre-sample mask has an area of $1.2 \times 15$ cm² with 21.4 μm large apertures extending along the 15 cm length with an average period 75 μm; apertures are arranged in a 4-way asymmetric configuration, in which their consecutive positions are shifted by −10 μm, 0 μm, 10 μm and 20 μm in a repeating pattern. Both masks were obtained by finely aligning and joining together three separate segments. Both masks sit on motor stacks that allow their alignment, namely two Newport (Irvine, CA) micro-translators oriented along $x$ and $z$ and two Kohzu (Kawasaki, Japan) goniometers for rotation around the $z$ and $x$ axis (see Fig. 5). An additional, larger Newport translator is used to scan the samples along the $x$ direction. All distances among the various components are given in Fig. 5(a).

### Phase retrieval
Phase retrieval in EI is based on the comparison between illumination curves (ICs) obtained with and without the sample present. The IC is the bell-shaped curve obtained when the detector mask is kept still, with the apertures placed at the centre of the respective detector pixels, the pre-sample mask is scanned in sub-period steps along the $x$ direction, and the transmitted x-ray intensity is recorded at each position (dashed line in Fig. 6(a)). The introduction of a sample causes three effects: a lateral shift, an increase in the full width at half maximum (FWHM), and a reduction in the area under the curve (solid line in Fig. 6(a)). These correspond to refraction, dark-field and attenuation signals, respectively.

Mathematically the IC can be expressed as the convolution of the two mask apertures with the (projected) focal spot distribution. Gaussian curves were proven to provide a good approximation for the IC shape[15], mostly thanks to the approximately Gaussian shape of the x-ray focal spot. Therefore, retrieval is performed by fitting Gaussians to ICs before and after the introduction of a sample, and comparing their centres, FWHM and areas to obtain refraction, dark-field, and attenuation, respectively—by subtraction in the first two cases and division in the third.

Fitting a Gaussian requires at least three parameters, therefore if both sample and detector masks have regular periods, three frames need to be acquired with different displacements of the sample mask along $x$, to sample three different points on the IC. If a scanning system is used, this could be replaced by the introduction of an asymmetric mask.

The concept is exemplified in Fig. 6 panels (b) and (c). Different aperture groups (1, 4, 7, 10...; 2, 5, 8...; 3, 6, 9... - colour-coded in red, blue, and green, respectively) sample different positions on the IC. Following an object scan, data from corresponding aperture groups are summed, forming images collected on the left-hand side, top and right-hand side of the IC. These are then fed to the retrieval algorithm discussed below.

For the experiments presented here, two measures were taken to obtain a more robust retrieval. First, a 4-way instead of a 3-way asymmetric mask was employed (Fig. 7(a)). Alongside stabilising the fit, this enables extracting an offset parameter since Gaussians from

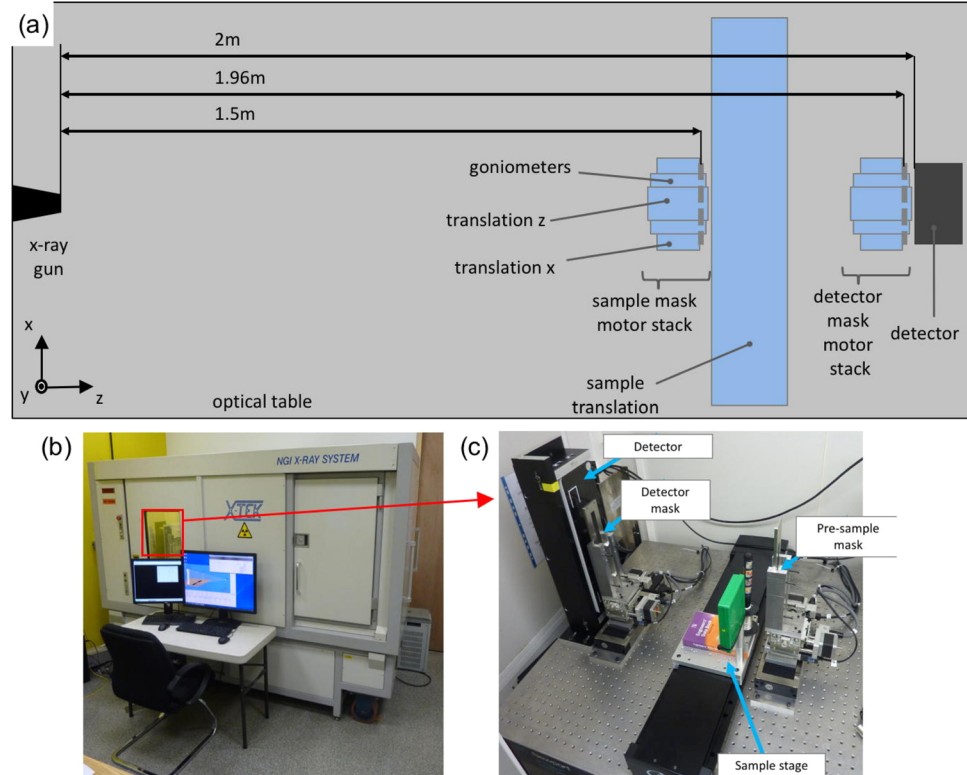

**Fig. 5 | Schematic and photos of the imaging system. a** shows a schematic of the imaging system, listing all components and their respective distances. All components are mounted on an optical table, placed inside the cabinet shown in **b**.

**c** shows part of the interior of the cabinet in which detector, detector and sample masks, and sample stage are visible. The x-ray source (not visible) is to the far right of the components shown in panel **c**.

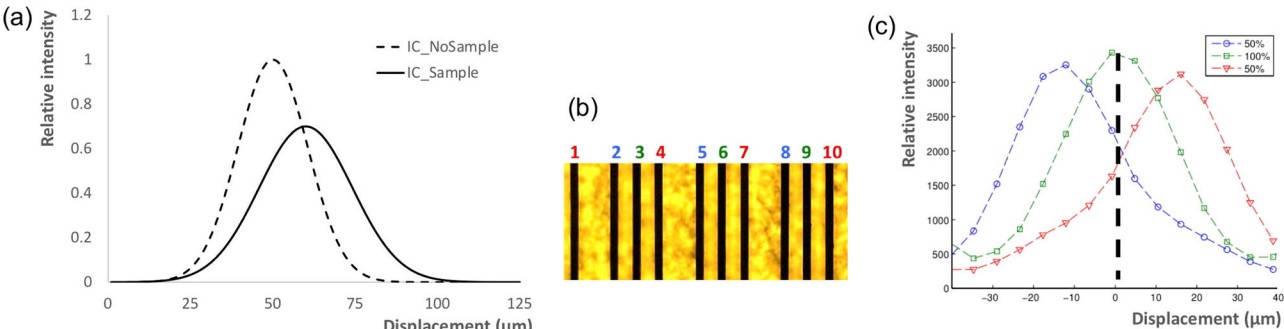

**Fig. 6 | Illumination curves and basic principles of the phase retrieval process. a** IC before (dashed line) and after (solid line) the introduction of a sample. **b** micrograph of a 3-way asymmetric sample mask, with the apertures in black visible on the yellow gold plating. Apertures 1, 4, 7, 10... are pulled back with respect to the regular period of apertures 3, 6, 9..., while apertures 2, 5, 8... are pushed forward. When such an asymmetric mask is scanned against a stationary

detector mask and x-ray intensities are recorded at every position, slightly shifted ICs are obtained for each group of apertures (panel **c**), note the colour coding matching the numbers in panel **b**. Hence when the pre-sample mask is placed at the position indicated by the thick dashed line in **c**, each aperture group samples a different position on the IC.

neighbouring apertures normally do not reach zero, due primarily to the residual transmission of the x-ray beam through the masks. Secondly, rather than fitting each Gaussian independently (as in Fig. 7(a)), groups of four Gaussian relating to 4 neighbouring pixels (Fig. 7(b)) were fitted with the following function:

$$IC_0 = \sum_{i=1}^{4} \frac{a_{i0}}{\sqrt{2\pi a_{i2}^2}} \exp\left[\frac{(x - a_{i1})^2}{2a_{i2}^2}\right] + a_3 \tag{1}$$

$IC_0$ is obtained from object-free scans, and as a result, it can be finely sampled off-line by placing the pre-sample mask in multiple positions. Therefore, parameters $a_{ij}$ can be estimated with a high degree of precision. When a real sample is scanned, four points such as those highlighted by the green dots in Fig. 7(b) are used to fit the following function:

$$IC = \sum_{i=1}^{4} \frac{a_{i0}t}{\sqrt{2\pi(a_{i2}^2 + \tau^2)}} \exp\left[\frac{(x - a_{i1} - \Delta)^2}{2(a_{i2}^2 + \tau^2)}\right] + a_3 o \tag{2}$$

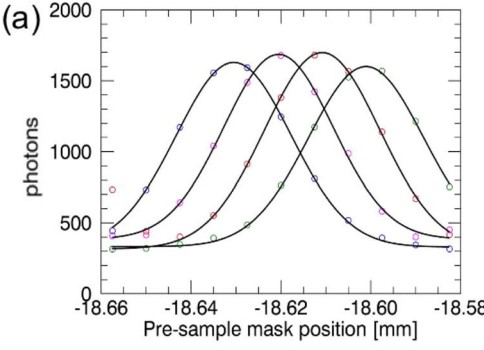

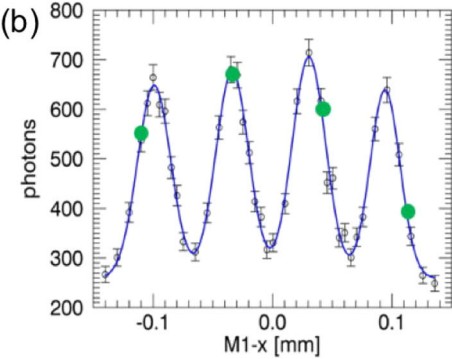

**Fig. 7 | Practical implementation of the phase retrieval process. a** ICs from a 4-way asymmetric pre-sample mask (experimental points and Gaussian fits are shown for each aperture, with different colour points corresponding to different apertures). **b** their fit as a single 4-Gaussian function rather than through four separate Gaussians. The black dots with error bars (corresponding to one standard deviation) represent a finely sampled version, the solid blue curve the 4-Gaussian fit, and the green dots represent typical sampling points in an object scan, determined by the relative positions of sample and detector masks. Note these correspond to a certain fixed position of the pre-sample mask.

From which parameters $t$, $\tau^2$ (proportional to the difference between squared full widths at half maximum of the beamlet with and without the sample, referred to as $\sigma^2 - \sigma_0^2$ in the text) and $\Delta$ can be obtained, corresponding to the sample's transmission, dark-field and refraction signals, respectively. The parameter $o$ is also extracted, which corresponds to an attenuation image of the sample obtained with the portion of the beam spectrum transmitted through the gold septa (offset image).

### Sample preparation

For the earliest results (Fig. 2(a), (b)), threat and non-threat materials were placed in acrylic containers with a fixed (5 mm) thickness. For the results of Fig. 2(c), (d), four separate containers with thicknesses 2.5 mm, 5 mm, 10 mm, and 20 mm were filled with each material, allowing to scan 15 different thicknesses ranging from 2.5 mm to 37.5 mm for every material. For the results of Fig. 3 and Supplementary Fig. 5, random thickness boxes with random content were placed inside bags containing other cluttering items, such as petroleum jelly, body puff, lip balm, toothbrushes, dental floss, tampons, a sewing kit, earphones, breath freshener, nail file, paracetamol, hairbrush, gum, and a charger. For the results of Fig. 4, varying quantities of C4 were placed inside a laptop, a mobile phone, and a hair-drier, either into existing gaps, or by creating ad hoc ones by removing components such as batteries, hard disk etc. The overall dataset for this second PoC test contained 576 scans, of which 80 were held-out for testing. Out of the remaining 496 image scans, the explosive/non-explosive split was kept nearly even, i.e., we had around 50% of the scans containing the explosive C4 and the rest containing no explosive. Approximately 80% of the images were used for training and the remaining 20% were used for validation. Adequate care was taken so that the validation and the training datasets were not skewed with respect to the number of image scans containing the explosives. Supplementary Fig. 7 provides a scheme listing all areas of concealment. This features images of the electrical items without any concealed explosive, but in which coloured areas indicate the various regions where C4 was concealed during the trial. Electrical items with and without concealed C4 were then placed in bags alongside a range of cluttering objects, as previously done to obtain the results of Fig. 3 and supplementary Fig. 5.

### Proof of concept CNN test 1: loss functions

As reported in the main article, we have evaluated three losses: softmax, cross-entropy, and hinge (max-margin), which are three of the most widely used loss functions for classification tasks in deep learning. With the softmax loss layer, the training of a CNN is typically accomplished by minimizing the following cost or error function (negative log-likelihood):

$$\mathscr{L}_s = -\frac{1}{N}\sum_{r=1}^{N}\log(p'_{r,y_r}) + \mathscr{L}_R \tag{3}$$

where $r$ indexes $N$ training images across all categories, $\mathscr{L}_R = \lambda\|W\|_2$ is the L2 regularization on the weights $W$ of the deep net, $\lambda$ is a regularization parameter, and the probability $p'_{r,y_r}$ is obtained by applying the softmax function to the $M$ outputs of the last layer (with $M$ the number of classes we wish to predict labels for), and $y_r$ is the output label for the $r^{th}$ image. Letting $l_{r,m}$ denote the $m^{th}$ output for the $r^{th}$ image, we have:

$$p'_{r,m} = \frac{e^{l_{r,m}}}{\sum_{m'}e^{l_{r,m'}}} \tag{4}$$

where $m$ and $m'$ index $M$. Using $p'_{r,m}$ we can also specify the hinge loss (a max-margin loss) as follows:

$$\mathscr{L}_h = -\frac{1}{N}\sum_{r=1}^{N}\sum_{m=1}^{M}\max(0, 1 - \delta(l_{r,m}=m)p'_{r,m}) + \mathscr{L}_R \tag{5}$$

where:

$$\delta(\text{condition}) = \begin{cases} 1 & \text{if condition} \\ -1 & \text{otherwise} \end{cases} \tag{6}$$

when the sigmoid cross entropy loss is applied, each image is expected to be annotated with a vector of ground-truth label probabilities $\mathbf{p}_r$, having length $M$, and the network is trained by minimizing the following loss objective:

$$\mathscr{L}_e = -\frac{1}{NM}\sum_{r=1}^{N}[\mathbf{p_r}\log\mathbf{p'_r} + (1 - \mathbf{p_r})\cdot\log(1 - \mathbf{p'_r})] + \mathscr{L}_R \tag{7}$$

where the probability vector $\mathbf{p}'_r$ is obtained by applying the sigmoid function to each of the $M$ outputs of the last layer.

A cursory comparison between the equations of hinge and cross-entropy loss may suggest that, while with cross-entropy loss learning would progress for increased separation, with hinge loss it may stop once the samples are immediately on the correct side of the margin. However, this is not the case, since in practice hinge loss is implemented slightly differently, with the learning continuing for maximal separation even when the samples come to the correct side of the margin[30]. In fact, due to the nature of the loss surfaces of hinge and

cross-entropy, hinge loss can keep trying to achieve separation at the same rate throughout the training, while cross-entropy would only look for good separation during some initial stages of learning and, if this is achieved, would then lessen its priority to further separate. Thus, when the features are subtle to segregate (as in our case), hinge loss can achieve better performance than cross-entropy.

Note also that, in our case, hinge outperforms softmax loss. With overlapping threat and non-threat materials, no two samples can be seen to be completely mutually-exclusive, hence segregation depends on how a given loss can separate appropriate attributes across the samples of the two classes when some attributes are similar. Due to its max-margin formulation, hinge loss is empirically seen to choose adequate attributes more correctly, in comparison to the log-likelihood maximization formulation in softmax loss.

Supplementary Figs. 8 and 9 provide schematics of our type 1 and type 2 architectures, respectively.

**Proof of concept CNN test 1: training details**
We specify below the parameter configuration used for training CNNs with all our deep learning procedures in PoC test 1:

**Optimizer:** ADAM
**Optimizer Parameter:** 0.99
**Weight Regularization Parameter:** $\lambda$ = 0.0001
**Regularization Type:** L2
**Initial Learning Rate:** 0.001
**Learning Decay Type:** Exponential
**Learning Decay Step Size:** 100
**Learning Decay Multiplication Factor:** 0.96
**Dropout Probability:** 0.50
**Crops during Inference:** 2
**Batch Size:** 32

All training and inference is performed on an NVIDIA GeForce 1080Ti GPU with around 11 GB of available of on-chip memory. All our deep learning experiments have been carried out using the TensorFlow library. In all cases, we train for 40 epochs.

**Proof of concept CNN test 2: architecture description**
Since in this case, the dataset consists of complex images featuring different electronic items that largely obscure the target material, a multistep process was adopted through a split network, where the electronic items are segregated first, then the target material is detected. A schematic of the split network is provided in Supplementary Fig. 10.

The architecture contains two layers, i.e., it can be seen as a stacked architecture. The first layer is targeted to segregate the compositions of electrical items as far as possible from one another, while the second layer aims to discriminate the textures of explosives against those of non-explosives. The second layer uses the dark-field and ratio images (which add information on the materials micro-structure) and the features learnt from the first layer. Since the explosives are typically occluded by other object parts, explosive textures are often an amalgamation of explosive characteristics with those of the containing/surrounding objects, which explains why pre-extracting cues from the object composition helps the task, and justifies the use of this stacked architecture.

GoogleNet[26] was used as the base architecture. While Inception ResNet gave the best results in PoC Test 1, it experienced intermittent convergence issues on this new dataset, while GoogleNet showed a consistent convergence trend. Since this provided sufficiently good results for the second PoC test, we did not delve deeper into the convergence issues with ResNet; this will be explored in future research, especially when dealing with more

diverse datasets. The use of three different networks in the second layer was also an empirical choice, which provided the best results. The entire network is trained end-to-end with a softmax loss function, using transfer learning, i.e., pre-train the network with a large dataset (ImageNet) and fine-tune only the last layer with our own dataset. With the stacked architecture, features at each level were easier to segregate, which we think explains why in this case the hinge loss did not show any clear advantage over softmax. Based on the results of PoC Test 1, we did not use any data augmentation. No image crops were used during training or testing, since the dataset size was comparable to PoC Test 1 without any split rounds. Finally, a texture network was not added in this case. In an initial ablation study, we applied the Type II architecture (Supplementary Fig. 9) to the new dataset, but failed to see a convergence trend, possibly due to complex classes each containing an explosive or a non-explosive, plus benign cluttering objects and electronic items. We thus resorted to a stacked architecture, with the first stack providing information on the electronic items and the second looking into the separation between explosive and non-threat objects. For the second stack, we tested varying numbers of CNNs, and obtained the best results with three. One viable approach would be to merge a texture recognition CNN with each component CNN; while the results obtained in PoC test 2 were considered sufficiently good as to make this unnecessary, this will be explored in future research on more complex datasets.

**Proof of concept CNN test 2: training details**
We specify below the parameter configurations used for training CNNs with all our deep learning procedures in PoC test 2:

**Optimizer:** MOMENTUM
**Optimizer Parameter:** 0.99
**Weight Regularization Parameter**: $\lambda$ = 0.0001
**Regularization Type:** L2
**Initial Learning Rate:** 0.001
**Learning Decay Type:** Exponential
**Learning Decay Step Size:** 200
**Learning Decay Multiplication Factor:** 0.92
**Dropout Probability:** 0.50
**Crops during Inference:** None
**Batch Size:** 32

**De visu trials**
To simplify the viewing procedure, it was decided to show only one (the attenuation) image to the operators. These were instructed in advance i.e., they were told they were about to see 80 images on a screen, that they had 20 min to complete the task, that all of these images contained an electrical item, and that a subset of these electrical items contained C4, but this was never placed *outside* the electrical items. They were also told that only C4 was present and not complete explosive devices, and therefore that they should not be looking for components such as detonators etc. They were given a mark sheet with clear/not clear checkboxes for each image. It is important to note that these operators would normally be presented with (and are trained for) colour-coded, dual-energy images where organic and inorganic materials are separated, rather than single-energy, grey scale images like in this case; one should therefore assume their performance would be significantly improved were they provided with the former.

Both times reported in the last column of Table 1 refer to the entire dataset of 80 images. For the CNN test, this includes timings for initial loading of the CNN architectures, data transfer to GPU local memory, data pre-processing, inference to predict the output probabilities on an NVIDIA GeForce GTX 1080Ti GPU, storing saliency map visualizations, and dumping of results in a text file.

## Data availability

The datasets generated and/or analysed during the current study are not publicly available on grounds of security but are available from the corresponding author on request.

## Code availability

Computer code used to perform phase retrieval is available from AA. Machine learning code is available from SSS. Interested readers are invited to contact the corresponding author.

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

## Acknowledgements

This work was supported by the Innovative Research Call in Explosives and Weapons Detection. This is a Cross-Government programme sponsored by a number of Departments and Agencies under the UK Government's CONTEST strategy in partnership with the US Department of Homeland Security, Science and Technology Directorate (grant 37765-271212; A.A., M.E., A.O.). Additional support was obtained from HMG's Defence and Security Accelerator (DASA) under the Future

Aviation Security Solutions (FASS) programme (grants ACC101705 and ACC106964; T.P., S.S.S., T.R.-S., A.O.) and from the Engineering and Physical Sciences Research Council (EPSRC) (grant EP/T005408/1; M.E., S.A., A.O.). M.E. was supported by the Royal Academy of Engineering under the RAEng Research Fellowships scheme (grant RF1415\14\33). A.O. was supported by the Royal Academy of Engineering under the Chairs in Emerging Technologies scheme (grant CiET1819/2/78). The authors would like to thank Prof Roberto Cipolla (Department of Engineering, University of Cambridge) for fruitful discussions.

## Author contributions

A.O., D.B., I.G.H., and T.R.-S. designed research. T.P., A.A., M.E. collected data. T.P., A.A., F.A.V., M.E., S.S.S., S.A., and A.O. analysed data. A.O. wrote the manuscript. All authors reviewed the manuscript.

## Competing interests

Bate, Haig and Astolfo are (or were at the time this research was carried out) Nikon employees; Nikon holds a license to exploit the EI technology in various areas excluding security. Riley-Smith is the CEO of XPCI Technology Ltd which aims at exploiting the EI technology in security. Olivo, Endrizzi and Vittoria are named inventors on patents held by UCL protecting the EI technology. Shankar is the CEO of Nylers Ltd and acts as DNN consultant to Riley-Smith. All other authors have no competing interests to disclose.

## Additional information

**Supplementary information** The online version contains

supplementary material available at https://doi.org/10.1038/s41467-022-32402-0.

