## [Peer Review File · Nature Communications]

Enhanced detection of threat materials by dark-field x-ray imaging combined with deep neural networksEditorial Note: This manuscript has been previously reviewed at another journal that is not operating a transparent peer review scheme. This document only contains reviewer comments and rebuttal letters for versions considered at Nature Communications.

REVIEWER COMMENTS

Reviewer #1 (Remarks to the Author):

The authors have satisfactorily addressed the concerns raised.

Reviewer #2 (Remarks to the Author):

The paper addresses the problem of materials discrimination within the broader domain of multi-energy / multi-spectral X-ray imaging. The key idea is that the inclusion of dark-field X-ray information, which creates a texture that is characteristic of the imaged material, in addition to X-ray attenuation information leads to improved material discrimination in the case of threat vs. non-threat materials in a security context.

This is a joint imaging/signal understanding problem within the broader domain of X-ray image understanding spanning both objects and materials as it pertains to the use of X-ray technologies for aviation and border security-like functions.

The main theoretical contribution of the paper is to show that dark-field information provides improved materials separation within the joint signal space of dark-field vs. attenuation for high-low energy X-ray imaging.

Experimentally the paper shows a proof of concept that the inclusion of X-ray dark-field information provides improved materials discrimination above that of attenuation alone. The experiments and supporting analysis, however, only partially the claims of the paper in terms of the key contribution in general (see later).

Overall this contribution is notable against the current state of the art on the basis that the use of dark-field X-ray information for materials discrimination has not hitherto been considered within the security imaging context.

With reference to earlier reviews, it is clear that a substantive effort has been made by the authors to improve the clarity and presentation of the manuscript since an earlier submission to Nature (main journal).

Strengths:

Key novel ideas: the paper presents a number of novel ideas that encompass the use of dark-field X-ray information in addition to X-ray attenuation for materials discrimination. These ideas are of very significant importance because effective and accurate materials discrimination in X-ray imaging is a key requirement of security screening for aviation and border security

A clearly stated formalized scientific hypothesis is stated and present (i.e. dark-field X-ray information improves X-ray materials discrimination).

Experimental validation: the paper presents moderately convincing experimental analysis across two proof of concept experimental such that they provide adequate support for the primary claims of the paper.

Theoretical validation: the paper presents significant additional validation with reference to an

extended study of varying materials under both regular (attenuation) and dark-field X-ray imaging together with their resulting separability in the resultant X-ray signal space (Figure 2).

Notably, the paper offers clear explanations and illustrations (although the clarity of some of the references labeling to the Figure 2 sub - subfigures could be improved), contributions that are clearly stated and validated, and new technical insight to this problem.

The supporting experiments exhibit a moderate to good degree of scientific rigor throughout and are supported by additional analytical validation.

Overall the work offers the potential for a significant practical impact within the specific problem domain it addresses and additionally much wider impact on the broader security imaging research community.

Weaknesses:

General: by the authors' volition, the work presents two proof of concept experiments over very limited datasets using standard "off the shelf" deep learning approaches. Any novelty lies not in the dark-field X-ray imaging but its combined use for this task with such deep learning approaches. However, as it appears earlier reviewers have already picked up on, the methodology for the deep learning part is weak and could still result in overfitting despite the methodological steps to prevent this.

Why not use a much simpler machine learning approach to validate at least some of the claims? (e.g. traditional approach, reduced complexity CNN).

Results: The text on p.9, below Table 1 ("...this resulted in a decrease of about 11% in precision and around 20% in overall accuracy. ") seems very important in the context of the key Table 1 result which is performance in cluttered electronics. Why is this result not:

- (a) in Table 1 as entries for CNN with/without dark-field information
- (b) highlighted in the abstract / intro/conclusions to say words to the effect of: ... under evaluation the inclusion of dark-field imaging resulted in a +20% increase in overall accuracy and a +11% increase in precision. (or similar but translated to the language of TP vs. FP etc).

With reference to the results of Table 1 and the text below - is it possible to add a ROC plot of TP vs. FP on the basis of the threshold value used for the output of the CNN ? (which as far as I can see is itself not stated anywhere explicitly)

Related work: The work is really only supported by a limited overview of related work in the field and it is unclear what the performance levels of existing approaches for materials discrimination in X-ray imaging are in general prior to this work. For example a number of both established and more recent reviews of the field either cover or at least touch on this topic (in chrono. order):

- Singh, Sameer, and Maneesha Singh. "Explosives detection systems (EDS) for aviation security." *Signal processing* 83.1 (2003): 31-55.
- Wells, K., and D. A. Bradley. "A review of X-ray explosives detection techniques for checked baggage." *Applied Radiation and Isotopes* 70.8 (2012): 1729-1746.
- Mouton, Andre, and Toby P. Breckon. "A review of automated image understanding within 3D baggage computed tomography security screening." *Journal of X-ray Science and Technology* 23.5 (2015): 531-555.
- Rogers, Thomas W., et al. "Automated x-ray image analysis for cargo security: Critical review and future promise." *Journal of X-ray science and technology* 25.1 (2017): 33-56.
- Mery, Domingo, Daniel Saavedra, and Mukesh Prasad. "X-ray baggage inspection with computer vision: A survey." *IEEE Access* 8 (2020): 145620-145633.
- Akcay, Samet, and Toby Breckon. "Towards automatic threat detection: A survey of advances of deep learning within X-ray security imaging." *Pattern Recognition* 122 (2022): 108245.

Overall the paper is well presented and someone who is interested in this topic and familiar with the current state of the art would gain something from the insight it offers. However, if one of its primary claims is "proof of concept" relating to machine learning approaches the paper structure should follow the conventions for that field (e.g. Intro. -> Related Work -> Methodology -> Evaluation/Results/Discussion -> Conclusion/Further Work). At the moment the structure of the paper seems somewhat disjointed with the conventions of work in this field and perhaps more aligned to that of regular physics papers.

My current view of this work is that it still requires at least a major revision prior to publication at this level.

I feel that this evaluation is primarily justified on the basis of the fact that some crucial details on related work and performance of this technique relative to that work is missing in addition to issues around the presentation of the results in Table 1. The authors are also advised to revise the structure of the paper to conform to that of non-physics experimental, machine learning/pattern recognition work in the field.

Partridge *et al.* “Enhanced detection of threat materials by dark-field x-ray imaging combined with deep neural networks” – **Response to reviewers**

We would like to thank the reviewers for the time spent assessing our manuscript, and for their overall positive evaluation. In the following, we report their comments and suggestions on a point-by-point basis, and list the changes introduced to the manuscript in order to address them.

Reviewer #1 (Remarks to the Author):

The authors have satisfactorily addressed the concerns raised.

Thanks for your appreciation of the amount of work we have invested into revising our original manuscript.

Reviewer #2 (Remarks to the Author):

We do not report here again the reviewer’s overall assessment and description of the manuscript’s strengths, for which we are highly appreciative, and focus only on the points that require a direct action from our side to improve the manuscript (with the exception of the following sentence: “*Notably, the paper offers clear explanations and illustrations (although the clarify of some of the references labeling to the Figure 2 sub - subfigures could be improved)*”: to address this, we have rewritten the caption of Fig. 2 in an attempt to make it clearer, with all new/modified parts in **red**). Please also note that while addressing the above comment we noticed that the horizontal axes of the graphs on the left hand side of Fig 2(c) were mislabelled (as “ μx ” instead of “ μz ”), and this has been corrected.

We are equally appreciative of the reviewer’s suggestions and feel that their implementation has led to a significantly improved manuscript. Please note that, for ease of visualisation, all newly introduced or modified parts are in **red** in the new version of the manuscript.

Why not use a much simpler machine learning approach to validate at least some of the claims? (e.g. traditional approach, reduced complexity CNN).

We have re-processed the data using two simpler, widely available machine learning approaches – GoogleNet and Inception ResNet. This has highlighted two important results, namely the explicit demonstration that our custom split architecture leads to a better overall outcome, and that inclusion of the dark-field signal leads to a better performance also when simpler architectures are used. This is now discussed at the end of the Results section, and the results themselves have been included as Supplementary Materials 6.

Results: The text on p.9, below Table 1 (“...this resulted in a decrease of about 11% in precision and around 20% in overall accuracy. ”) seems very important in the context of the key Table 1 result which is performance in cluttered electronics. Why is this result not: - (a) in Table 1 as entries for CNN with/without dark-field information

- (b) highlighted in the abstract / intro/conclusions to say words to the effect of: ... under evaluation the inclusion of dark-field imaging resulted in a +20% increase in overall accuracy and a +11% increase in precision. (or similar but translated to the language of TP vs. FP etc).

Thanks for this suggestion. Table 1 has been expanded to include the results obtained while excluding the dark field signal, and this result is now mentioned both in the abstract (“**In both cases, application of the same approaches to datasets from which the dark-field images were removed leads to a clear degradation in performance**”) and in the introduction (“**e.g, in the second experiment, a reduction of about 11% in true positives and an increase of about 12% in false positives is observed, see table 1 below**”). We have not repeated this point in the conclusions though, because the sentence “Importantly, in both cases the performance was negatively affected if the process was repeated using only the multi-energy attenuation x-ray images (a “proxy” for current practice) and excluding the dark-field ones” was already present in the original version of the manuscript.

With reference to the results of Table 1 and the text below - is it possible to add a ROC plot of TP vs. FP on the basis of the threshold value used for the output of the CNN ? (which as far as I can see is itself not stated anywhere explicitly)

We apologise for this omission; the threshold value (0.5) is now explicitly stated (end of the results section); furthermore, a ROC curve was produced and added to the manuscript (as Supplementary Materials 7); this is also mentioned in the manuscript, again at the end of the results section.

Related work: The work is really only supported by a limited overview of related work in the field and it is unclear what the performance levels of existing approaches for materials discrimination in X-ray imaging are in general prior to this work. For example a number of both established and more recent reviews of the field either cover or at least touch on this topic (in chrono. order):

- Singh, Sameer, and Maneesha Singh. "Explosives detection systems (EDS) for aviation security." *Signal processing* 83.1 (2003): 31-55.
- Wells, K., and D. A. Bradley. "A review of X-ray explosives detection techniques for checked baggage." *Applied Radiation and Isotopes* 70.8 (2012): 1729-1746.
- Mouton, Andre, and Toby P. Breckon. "A review of automated image understanding within 3D baggage computed tomography security screening." *Journal of X-ray Science and Technology* 23.5 (2015): 531-555.
- Rogers, Thomas W., et al. "Automated x-ray image analysis for cargo security: Critical review and future promise." *Journal of X-ray science and technology* 25.1 (2017): 33-56.
- Mery, Domingo, Daniel Saavedra, and Mukesh Prasad. "X-ray baggage inspection with computer vision: A survey." *IEEE Access* 8 (2020): 145620-145633.
- Akcay, Samet, and Toby Breckon. "Towards automatic threat detection: A survey of advances of deep learning within X-ray security imaging." *Pattern Recognition* 122 (2022): 108245.

Overall the paper is well presented and someone who is interested in this topic and familiar with the current state of the art would gain something from the insight it offers. However, if one of its primary claims is "proof of concept" relating to machine learning approaches the paper structure should follow the conventions for that field (e.g. Intro. -> Related Work ->

Methodology -> Evaluation/Results/Discussion -> Conclusion/Further Work). At the moment the structure of the paper seems somewhat disjointed with the conventions of work in this field and perhaps more aligned to that of regular physics papers.

We agree with the reviewer and have introduced a “related work” section as a result. Please note that, as requested by the editor, we have not changed the overall structure of the paper, as our original submission followed the style of Nature Communications (with methods at the end etc). However, the “related work” section has been introduced immediately after the introduction, as requested by the reviewer (see text in red at page 3). This section includes all references suggested by the reviewer, plus one of the early examples of application of deep learning to security imagery. The reference list has been lengthened accordingly, with all new references in red.

My current view of this work is that it still requires at least a major revision prior to publication at this level.

We hope the above has addressed all of the reviewer’s comments and remain at his/her disposal for any additional request.

REVIEWERS' COMMENTS

Reviewer #2 (Remarks to the Author):

Thank you for addressing all of my comments effectively.